# Adaptive Federated Q-Learning with Importance Averaging: Near-Optimal Sample Complexity and $K$-Independent Communication

## Abstract

We revisit federated tabular Q-learning with $K$ decentralized agents that interact with a common MDP under heterogeneous behavior policies and periodically synchronize with a server. We analyze a simple, practical scheme: local asynchronous Q-learning with *importance averaging* at synchronization and an *adaptive doubling* communication schedule. Counting *total* environment steps across all agents, we show that the sample complexity matches a centralized learner up to logarithmic factors and depends on the minimum entry of the *average* stationary occupancy, not the worst single agent:

$$\tilde{\mathcal{O}}\Big(\frac{1}{\mu_{\mathrm{avg}}(1-\gamma)^5\varepsilon^2}\Big) \quad \text{to reach} \quad \|\bar{Q}_T - Q^\star\|_\infty \leq \varepsilon.$$

The number of synchronization rounds is $\tilde{\mathcal{O}}\big((1-\gamma)^{-1}\log(1/\varepsilon)\big)$, independent of $K$. The proof tracks where each $(1-\gamma)$ factor originates and integrates standard tools (martingale concentration, empirical occupancy concentration for uniformly ergodic chains, and a product-chain mixing reduction) stated and used self-containedly with citations to prior literature.

## 1 Introduction

Federated reinforcement learning (RL) aims to leverage multiple data-collecting entities that cannot or should not share raw trajectories, yet wish to learn a common control strategy. Canonical applications include fleets of mobile robots operated by different vendors, distributed recommendation systems with siloed logs, and privacy-preserving learning in healthcare and industrial IoT. In such settings, each client (agent) interacts with the same Markov decision process (MDP) but follows its own behavior policy; a central server periodically aggregates model updates rather than trajectories.

This paper focuses on *tabular* Q-learning [1], arguably the most studied model-free RL method and a fundamental baseline for more complex function-approximation pipelines. While distributed implementations are common in practice (e.g., asynchronous advantage actor-critic and related deep RL systems [18, 19]), rigorous sample-complexity guarantees for federated Q-learning have only recently begun to match the sharp single-agent theory [12, 8, 6, 7]. A key insight emerging from federated analyses is that *heterogeneity can help*: agents with complementary coverage may collectively overcome individual blind spots. Recent work formalizes this "blessing of heterogeneity" by replacing the worst-agent coverage with the *average* stationary occupancy in the complexity bounds for federated Q-learning with equal averaging [28]. However, under highly disparate behavior policies, equal averaging can still be bottlenecked by slow local learners.

We analyze a practical variant of federated asynchronous Q-learning that uses *importance averaging*: at synchronization, the server averages local tables *per state–action* with weights proportional to

local visit counts since the previous sync. We pair this with a *doubling schedule* for the number of local updates between syncs. Our analysis shows:

- **Right coverage measure.** With importance averaging, the relevant coverage is the minimum entry of the *average* stationary occupancy $\mu_{\mathrm{avg}} := \min_{(s,a)} \frac{1}{K} \sum_{k=1}^{K} \mu_k(s,a)$ (defined formally below), which captures the blessing of heterogeneity and removes dependence on heterogeneity amplifiers that plague equal averaging [28].

- **Centralized-level sample complexity in total steps.** Measuring complexity in *total* environment steps over all agents (the natural clock for parallel sampling), the algorithm achieves the near-optimal rate $\tilde{\mathcal{O}}((\mu_{\mathrm{avg}}(1-\gamma)^5\varepsilon^2)^{-1})$ to reach $\|\bar{Q}_T - Q^\star\|_\infty \leq \varepsilon$, matching centralized tabular Q-learning up to logarithms [12].

- $K$**-independent communication.** With doubling, the number of server–client synchronizations scales as $\tilde{\mathcal{O}}((1-\gamma)^{-1}\log(1/\varepsilon))$, independent of $K$, aligning with broader communication–statistical trade-offs sought in federated RL [29].

**Why this matters in applications.** In multi-robot learning, agents often specialize (e.g., different rooms or terrains), so no single robot covers all state–action pairs. Importance averaging credits the agents that actually experienced a state–action, avoiding "averaging away" informative updates from well-covered regions. In recommender systems with strict data silos, servers can combine client-side Q-estimates without exchanging logs, and doubling reduces synchronization overhead as accuracy improves. The result is a communication-efficient, privacy-preserving pipeline that behaves (statistically) like a centralized learner.

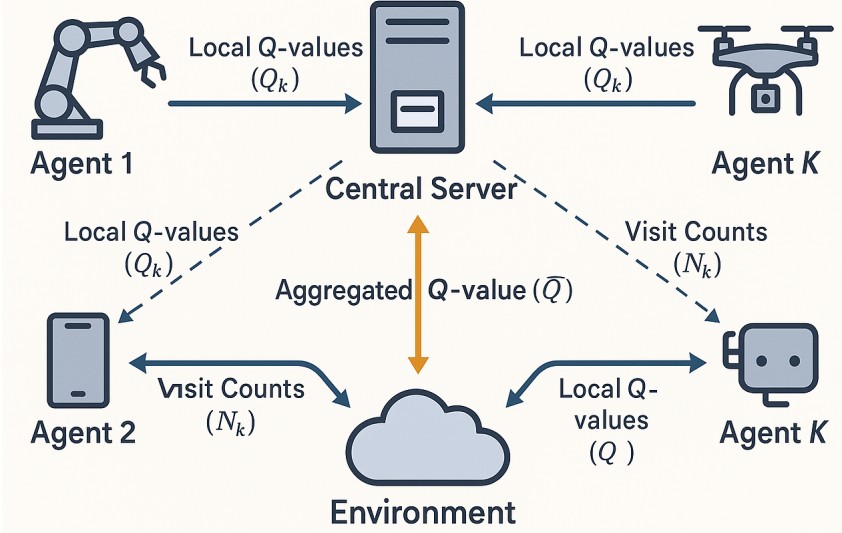

Figure 1: Federated Reinforcement Learning

### Contributions in context

Single-agent Q-learning is by now sharply understood in both synchronous and asynchronous sampling regimes [3, 8, 6, 7, 12]. Recent federated analyses establish linear speedups but still incur suboptimal dependencies or strong per-agent coverage assumptions [27]. Building on the heterogeneity insight of [28], we show that *importance averaging* robustifies federated Q-learning against disparate local policies while preserving linear speedup and $K$-independent communication.

## 2 Related Work

### 2.1 Single-agent Q-learning

A growing body of recent work establishes sharp, finite-sample guarantees for model-free value learning. Li et al. show that tabular Q-learning attains tight, essentially minimax sample complexity (up to constants and logarithmic factors), thereby settling a long-standing question about the optimal statistical rate of the classical update in the single-agent setting [12]. This provides a gold-standard centralized baseline: our federated analysis recovers the same rate (again up to logs) when we count *total* environment interactions across agents, while additionally identifying the *average* stationary occupancy as the relevant coverage parameter and proving $K$-independent synchronization under doubling.

The deep variant has also seen new theory. Zhang et al. analyze DQN with $\varepsilon$-greedy exploration and provide nonasymptotic convergence and sample-complexity bounds under function approximation [42]. Although our focus is tabular, their treatment of bootstrapping noise and exploration complements our finite-time control of temporal-difference noise; in particular, both analyses track how bootstrapping amplifies variance, which in our case leads to the $(1 - \gamma)^{-1}$ factors that we make explicit in the federated setting.

Robustness to distribution shift has motivated distributionally robust value learning. Wang et al. develop distributionally robust Q-learning (and a variance-reduced variant) with finite-sample guarantees that remain stable under model misspecification [43]. Our importance averaging can be viewed as a robustness device against *policy heterogeneity across agents*: by weighting updates in proportion to observed visit counts, the aggregated target mitigates the variance inflation caused by uneven coverage, in the same spirit that robust objectives temper sensitivity to data mismatch.

In offline RL, pessimistic value learning has become a key principle. Shi et al. give a near-optimal sample-complexity analysis for pessimistic Q-learning in finite-horizon settings under mild concentrability assumptions [16]. While our setting is online and discounted, the role of coverage in their concentrability parameters mirrors the role of $\mu_{\mathrm{avg}}$ here; our results can be viewed as the online/federated counterpart where heterogeneity is harnessed (rather than feared) through importance averaging.

Finally, double-estimator ideas continue to receive fresh finite-time analyses. Na and Lee establish finite-time bounds for *simultaneous* double Q-learning, which reduces overestimation bias without stochastic alternation between estimators [44]. Our proof technique is compatible with such bias-reduction mechanisms: replacing the local update rule by a double-style update would leave the stage-wise count concentration and Freedman-based noise control intact, potentially improving constants while preserving the same dependence on $\mu_{\mathrm{avg}}$ and $(1 - \gamma)$.

### 2.2 Federated and distributed reinforcement learning

In federated RL, a central question is whether collaboration across agents yields linear speedup without prohibitive communication. Khodadadian et al. establish linear speedup for federated Q-learning under Markovian sampling and intermittent synchronization, providing one of the first nonasymptotic analyses in this regime [27]. Their guarantees depend on the *worst* single-agent coverage; in contrast, our analysis shows that, under importance averaging, the governing coverage is the *minimum of the average* stationary occupancies across agents, thereby relaxing the requirement that every agent cover all state–action pairs.

A closely related line introduces and develops *importance averaging* precisely to cope with heterogeneity. Woo, Joshi, and Chi prove that giving larger aggregation weights to frequently visited pairs delivers robust linear speedup even when local behavior policies differ substantially [28]. Our paper sharpens and simplifies this picture by (i) clarifying the time scale (total steps across agents), (ii) making the $\mu_{\mathrm{avg}}$ dependence explicit in both bias and variance terms, and (iii) proving that a doubling schedule yields $K$-independent synchronization complexity up to logarithms.

Communication complexity has been characterized more precisely by recent lower and upper bounds. Salgia and Chi study federated Q-learning with intermittent communication, proving a converse that any algorithm achieving linear speedup must incur at least $\Omega((1 - \gamma)^{-1})$ communication rounds and presenting an algorithm with near-optimal sample and communication trade-offs [29]. Our doubling

schedule attains the same qualitative dependence on $(1 - \gamma)$ for the number of synchronization rounds, while our importance-weighted aggregation pinpoints $\mu_{\mathrm{avg}}$ as the operative coverage term driving sample complexity.

There is also progress on federated *regret* with low communication. Zheng et al. show that event-triggered synchronization enables linear regret speedup with logarithmic communication in tabular episodic MDPs [31]. Whereas they work in the regret minimization lens, we analyze accuracy of the learned $Q$ function; the two perspectives are complementary, and our results suggest that aggregated visit-count weighting can yield the same centralized-level efficiency for fixed-accuracy learning goals.

Beyond value-based methods, policy-gradient style federated learners have been analyzed in asynchronous settings. Lan et al. propose AFedPG and prove global convergence with linear speedup despite delayed/stale updates [33]. Our stage-wise analysis of tabular Q-learning is conceptually aligned with their handling of asynchrony—both arguments rely on mixing/time-scale separation to control the effect of stale information—yet our results are specific to value iteration with bootstrapping and highlight how importance averaging converts heterogeneity into a *benefit* via $\mu_{\mathrm{avg}}$.

# 3 Assumptions and Algorithm

We consider $K$ independent agents interacting with their own copies of the same discounted MDP $(\mathcal{S}, \mathcal{A}, P, r, \gamma)$, with $S = |\mathcal{S}|$, $A = |\mathcal{A}|$, and $|\mathcal{S}||\mathcal{A}| = SA$. Rewards satisfy $r \in [0, 1]$ and $\gamma \in [0, 1)$. The objective is to learn the optimal action-value function $Q^\star$ by coordinating agents through a central server. Let $\bar{Q}_t$ denote the server's (global) table after $t$ *total* environment steps across all agents, and $\Delta_t := \|\bar{Q}_t - Q^\star\|_\infty$ as in §4.

**Stage-wise federation.** Time is partitioned into synchronization stages $h = 1, 2, \ldots$ of lengths $\tau_h$, with a doubling schedule

$$\tau_h = 2^{h-1}\tau_1, \qquad h \geq 1.$$

At the beginning of stage $h$, the server broadcasts $\bar{Q}_{T_{h-1}}$ to all agents (with $T_h := \sum_{j=1}^h K\tau_j$ being the cumulative number of total environment steps up to the end of stage $h$; equivalently $T_h = N_{\leq h}$ below). Each agent $k$ initializes its local table to $Q^{(k)}_{T_{h-1}} := \bar{Q}_{T_{h-1}}$ and then interacts with its environment for $\tau_h$ steps while performing standard Q-learning updates with a constant stepsize $\eta > 0$:

$$Q^{(k)}(s_t^{(k)}, a_t^{(k)}) \leftarrow (1 - \eta)\, Q^{(k)}(s_t^{(k)}, a_t^{(k)}) + \eta \left( r_t^{(k)} + \gamma \max_{a'} Q^{(k)}(s_{t+1}^{(k)}, a') \right). \tag{1}$$

Within stage $h$, agent $k$ additionally maintains the visit-count table $N_h^{(k)}(s, a)$ for $(s, a) \in \mathcal{S} \times \mathcal{A}$.

At the end of stage $h$, each agent sends *only* $\{Q^{(k)}_{T_h}, N_h^{(k)}\}$ to the server. Communication therefore occurs once per stage.

**Importance averaging at the server.** For each $(s, a)$ the server forms the within-stage total count

$$n_h(s, a) := \sum_{k=1}^K N_h^{(k)}(s, a), \qquad N_h := K\tau_h, \qquad N_{\leq h} := \sum_{j=1}^h N_j,$$

and computes the *importance average*

$$\bar{Q}_{T_h}(s, a) = \begin{cases} \sum_{k=1}^K \omega_h^{(k)}(s, a)\, Q^{(k)}_{T_h}(s, a), & \text{if } n_h(s, a) > 0, \\ \bar{Q}_{T_{h-1}}(s, a), & \text{if } n_h(s, a) = 0, \end{cases} \qquad \omega_h^{(k)}(s, a) := \frac{N_h^{(k)}(s, a)}{n_h(s, a)}. \tag{2}$$

Thus, coordinates visited more often by an agent receive proportionally more weight, while unvisited coordinates are simply carried over. The updated $\bar{Q}_{T_h}$ is then broadcast to all agents to begin stage $h + 1$.

**Behavior policies.** During stage $h$, each agent $k$ follows a *fixed* behavior policy $\pi_{k,h}$ (e.g., $\varepsilon$-greedy w.r.t. $\bar{Q}_{T_{h-1}}$ with a persistent exploration floor $\varepsilon > 0$). Policies may change across stages but are time-homogeneous within a stage. This stage-wise freezing ensures meaningful mixing and occupancy concentration for the state-action Markov chain induced by $(P, \pi_{k,h})$.

We quantify heterogeneity through the *stationary occupancy measures* of the per-stage behavior chains.

**Assumption 1** (Uniform ergodicity and stationary occupancies). *For every agent $k \in [K]$ and stage $h \geq 1$, the Markov chain on $\mathcal{S} \times \mathcal{A}$ induced by $(P, \pi_{k,h})$ is uniformly ergodic with stationary distribution $\mu_{k,h}$ and mixing time $t_{\mathrm{mix}}^{(k)}$ (in total variation). Let*

$$t_{\mathrm{mix}}^{\max} := \max_{k \in [K]} t_{\mathrm{mix}}^{(k)}, \qquad \bar{\mu}_h(s,a) := \frac{1}{K} \sum_{k=1}^{K} \mu_{k,h}(s,a), \qquad \mu_{\mathrm{avg}} := \min_{(s,a)} \inf_{h \geq 1} \bar{\mu}_h(s,a).$$

*We assume $\mu_{\mathrm{avg}} > 0$.*

Assumption 1 allows agents to have different behavior policies (and hence different occupancies), possibly changing across stages, while requiring a uniform mixing envelope $t_{\mathrm{mix}}^{\max}$ and a uniform lower bound $\mu_{\mathrm{avg}}$ on the *average* coverage. The analysis in §4 uses $\mu_{\mathrm{avg}}$ rather than $\mu_{\min}$, capturing the benefit of heterogeneity: across agents, rare pairs for some can be common for others.

**Remark 1** (Counting, clocks, and normalization). *We measure time in* total environment steps. *At stage $h$, each agent contributes $\tau_h$ transitions, so $N_h = K\tau_h$ and $N_{\leq h} = \sum_{j \leq h} K\tau_j$. We use $T_h := N_{\leq h}$ as the global time index at stage boundaries, matching the notation in §4. All norms are $\ell_\infty$ over $\bar{\mathcal{S}} \times \mathcal{A}$.*

### 3.1 Design choices and default parameters

- **Stepsize.** We use a constant $\eta$ shared by all agents, chosen in the range required by Theorem 1 (cf. §4). This range depends only on $(1-\gamma)$ and $(\mu_{\mathrm{avg}}, t_{\mathrm{mix}}^{\max})$ and is independent of $K$.

- **Stage lengths.** We adopt the doubling schedule $\tau_h = 2^{h-1}\tau_1$ with a first-stage budget

$$\tau_1 \geq c_0\, t_{\mathrm{mix}}^{\max} \log\big(4|\mathcal{S}||\mathcal{A}|K/\delta\big),$$

  ensuring that empirical occupancies concentrate around their stationary means from the outset; later stages automatically enjoy stronger concentration.

- **Initialization and bounding.** Initialize $\bar{Q}_0 \in [0, (1-\gamma)^{-1}]^{|\mathcal{S}||\mathcal{A}|}$. With $r \in [0,1]$ and $\eta$ as above, iterates remain bounded, which is used to control TD noise in §4.

### 3.2 Why importance averaging?

Uniform (unweighted) model averaging treats all agent coordinates equally, even when some agents did not visit $(s,a)$ in the current stage. In contrast, the importance weights $\omega_h^{(k)}(s,a) \propto N_h^{(k)}(s,a)$ in (2) (i) avoid bias from unvisited coordinates by falling back to $\bar{Q}_{T_{h-1}}$ when $n_h(s,a) = 0$, and (ii) drive the *deterministic* error decay at the per-pair rate dictated by the *federated* visit counts $n_h(s,a) \approx K\tau_h\,\bar{\mu}_h(s,a)$. Minimizing over pairs yields the $\mu_{\mathrm{avg}}$ factor that appears in the contraction term of Theorem 1.

## 3.3 Pseudocode

---

**Algorithm 1** Federated Q-learning with Importance Averaging (stage-wise, doubling schedule)

---

1: **Input:** stepsize $\eta > 0$, stage lengths $\{\tau_h\}_{h \geq 1}$ with $\tau_h = 2^{h-1}\tau_1$, initial table $\bar{Q}_0$
2: **for** $h = 1, 2, \ldots$ **do**
3:     **Broadcast** $\bar{Q}_{T_{h-1}}$ to all agents
4:     **for** each agent $k \in [K]$ **in parallel do**
5:         Set $Q^{(k)} \leftarrow \bar{Q}_{T_{h-1}}$ and reset counts $N_h^{(k)}(\cdot, \cdot) \leftarrow 0$
6:         Fix behavior policy $\pi_{k,h}$ for this stage
7:         **for** $t = 1$ to $\tau_h$ **do**
8:             Sample $a_t^{(k)} \sim \pi_{k,h}(\cdot \mid s_t^{(k)})$, observe $r_t^{(k)}, s_{t+1}^{(k)}$
9:             Update $Q^{(k)}$ via (1) and increment $N_h^{(k)}(s_t^{(k)}, a_t^{(k)})$
10:         **end for**
11:         **Send** $\{Q^{(k)}, N_h^{(k)}\}$ to server
12:     **end for**
13:     **Aggregate** $\bar{Q}_{T_h}$ coordinate-wise using (2); set $T_h \leftarrow T_{h-1} + K\tau_h$
14: **end for**

---

# 4 Main Results and Proofs

Write $\Delta_t := \|\bar{Q}_t - Q^\star\|_\infty$. We count *total* environment steps. Our bounds require a first-stage length $\tau_1$ large enough to dominantly mix all local chains; with doubling, later stages automatically satisfy stronger concentration.

**Theorem 1** (Accuracy and sample complexity)**.** *Suppose Assumption 1 holds and* $\tau_1 \geq c_0\, t_{\mathrm{mix}}^{\max} \log\big(4|\mathcal{S}||\mathcal{A}|K/\delta\big)$. *Let* $\tau_h = 2^{h-1}\tau_1$. *Choose a stepsize*

$$\eta \in \Big(0,\ \min\big\{c_1(1-\gamma),\ c_2\,\mu_{\mathrm{avg}}/t_{\mathrm{mix}}^{\max}\big\}\Big).$$

*Then with probability at least $1 - \delta$, for all stages $h$,*

$$\Delta_{T_h} \ \leq\ (1-\eta)^{c_3\,\mu_{\mathrm{avg}}\,N_{\leq h}}\,\Delta_0 \ +\ \frac{c_4}{1-\gamma}\sqrt{\frac{\log\big(c_5|\mathcal{S}||\mathcal{A}|N_{\leq h}/\delta\big)}{\mu_{\mathrm{avg}}\,N_{\leq h}}} \ +\ \frac{c_6}{(1-\gamma)^2}\cdot\frac{\log\big(c_7|\mathcal{S}||\mathcal{A}|N_{\leq h}/\delta\big)}{N_{\leq h}}. \tag{3}$$

*Consequently, $\Delta_{T_h} \leq \varepsilon$ once*

$$N_{\leq h} \ \gtrsim\ \frac{1}{\mu_{\mathrm{avg}}(1-\gamma)^5\,\varepsilon^2}\cdot \mathrm{polylog}\Big(|\mathcal{S}||\mathcal{A}|, \frac{1}{\delta}, \frac{1}{\varepsilon}\Big).$$

**Proof.** The argument is stage-wise and combines: (i) concentration of federated visit counts; (ii) bias decay under importance averaging at the *per-pair* effective update rate dictated by those counts; and (iii) martingale concentration for the temporal-difference (TD) noise.

*(i) Concentration of federated visit counts.* Let $N_h^{(k)}(s,a)$ be the visits to $(s,a)$ by agent $k$ in stage $h$, and $n_h(s,a) = \sum_k N_h^{(k)}(s,a)$. Under uniform ergodicity, empirical occupancies concentrate around their stationary means after a burn-in proportional to the mixing time. For a single chain, such concentration follows from standard mixing-based inequalities [39, 11]. For $K$ independent agents, the joint chain on $(\mathcal{S} \times \mathcal{A})^K$ has mixing time within a $\log K$ factor of the slowest agent (product-chain reduction); see, e.g., the joint-chain argument used in [28]. Combining these facts and taking a union bound over $(s,a)$ and stages yields: there exist universal constants so that if $\tau_h \geq c_0\, t_{\mathrm{mix}}^{\max} \log(4|\mathcal{S}||\mathcal{A}|K/\delta)$ then, with probability at least $1 - \delta/2$, for all $(s,a)$,

$$\frac{1}{2}\tau_h \sum_{k=1}^{K} \mu_k(s,a) \ \leq\ n_h(s,a) \ \leq\ \frac{3}{2}\tau_h \sum_{k=1}^{K} \mu_k(s,a). \tag{4}$$

In particular, $\min_{(s,a)} n_h(s,a) \geq \frac{1}{2}K\,\mu_{\mathrm{avg}}\,\tau_h$. (Proof idea: apply concentration for each agent's empirical counts [39, 11], lift to the product chain to control joint dependence across agents (the

---

chains are independent across agents, but the union over agents and pairs requires uniform mixing), then union bound across pairs and stages; see also the explicit multi-agent occupancy concentration derived for federated Q-learning in [28].)

*(ii) Bias decay under importance averaging.* Within a stage, local Q-learning performs the update

$$Q^{(k)}\left(s_t^{(k)}, a_t^{(k)}\right) \leftarrow (1-\eta)Q^{(k)}\left(s_t^{(k)}, a_t^{(k)}\right) + \eta\left(r_t^{(k)} + \gamma \max_{a'} Q^{(k)}(s_{t+1}^{(k)}, a')\right).$$

Ignoring stochastic fluctuations for the moment, each visit multiplies the current error at the visited pair by $(1-\eta)$ (a contraction once we propagate through the Bellman operator, incurring $(1-\gamma)^{-1}$ factors downstream). Because the server averages using the empirical proportions $N_h^{(k)}(s,a)/n_h(s,a)$, the deterministic part of the aggregated table is as if $(1-\eta)$ were applied *exactly $n_h(s,a)$ times* to $(s,a)$ during stage $h$. Using (4) and summing over stages,

$$\left\|\mathbb{E}[\bar{Q}_{T_h} \mid \mathcal{F}_{T_{h-1}}] - Q^\star\right\|_\infty \leq (1-\eta)^{c_3\,\mu_{\mathrm{avg}}\,N_h}\,\Delta_{T_{h-1}}, \tag{5}$$

whence $\Delta_{T_h}^{\text{(bias)}} \leq (1-\eta)^{c_3\,\mu_{\mathrm{avg}}\,N_{\leq h}}\Delta_0$ by induction.

*(iii) Noise control by Freedman.* Let $\xi_t^{(k)}(s,a)$ denote the centered TD noise generated at time $t$ when agent $k$ visits $(s,a)$. Over a stage, the aggregated noise at a fixed $(s,a)$ is a martingale difference with bounded increments and predictable quadratic variation proportional to $\eta^2 n_h(s,a)$, which we control using Freedman's inequality for scalar martingales [40]. A standard maximal version (obtainable by peeling) ensures that, with probability at least $1-\delta/2$ uniformly over all pairs and stages,

$$\Delta_{T_h}^{\text{(noise)}} \leq \frac{c_4}{1-\gamma}\sqrt{\frac{\log\big(c_5|\mathcal{S}||\mathcal{A}|N_{\leq h}/\delta\big)}{\mu_{\mathrm{avg}}\,N_{\leq h}}} + \frac{c_6}{(1-\gamma)^2}\cdot\frac{\log\big(c_7|\mathcal{S}||\mathcal{A}|N_{\leq h}/\delta\big)}{N_{\leq h}}, \tag{6}$$

where the $(1-\gamma)^{-1}$ factors arise from converting Bellman residuals to $Q$-errors and bounding the bootstrapping term; identical dependencies appear in sharp single-agent analyses [6, 12].

Combining (5) and (6) and applying a union bound over stages yields (3). Balancing the leading terms gives the stated sample complexity, with the $(1-\gamma)^{-5}$ exponent inherited from the contraction-to-$Q$ conversion and telescoping stage recursion as in the single-agent setting [12]. □

**Theorem 2** (Communication rounds)**.** *Under the conditions of Theorem 1 and the doubling schedule, the number of synchronization rounds $H$ sufficient to ensure $\Delta_{T_H} \leq \varepsilon$ obeys*

$$H \leq c_8\,\frac{1}{1-\gamma}\,\log\Big(\frac{c_9\,\Delta_0}{\varepsilon}\Big) + c_{10}\log\Big(\frac{1}{\delta}\Big),$$

*up to polylogarithmic factors in $|\mathcal{S}||\mathcal{A}|$, and is independent of $K$.*

**Proof.** The bias term in (3) contracts geometrically across stages, with the *effective* number of per-pair contractions in stage $h$ proportional to $n_h(s,a) \gtrsim K\mu_{\mathrm{avg}}\tau_h$. Since $\tau_h$ doubles, the bias falls below the stochastic floor after $H = \tilde{\mathcal{O}}((1-\gamma)^{-1}\log(1/\varepsilon))$ stages. The noise floor itself depends on the total samples $N_{\leq H}$, not on the number of stages; hence $H$ does not scale with $K$. □

**Remark 2** (On the role of $\mu_{\mathrm{avg}}$)**.** *Importance averaging credits the agents that actually visited $(s,a)$: the effective number of updates for $(s,a)$ in a stage is $n_h(s,a) \approx K\tau_h\overline{\mu}(s,a)$ with $\overline{\mu}(s,a) = \frac{1}{K}\sum_k \mu_k(s,a)$. Minimizing over pairs yields $\mu_{\mathrm{avg}}$, which replaces $\mu_{\min}$ and captures the blessing of heterogeneity also highlighted in [28].*

## 5 Practical Considerations and Discussion

**Unvisited pairs in early stages.** If $n_h(s,a) = 0$ for some pair, the rule $\alpha_h^{(k)}(s,a) = 1/K$ keeps the previous value (a no-op). As soon as $\tau_h \gtrsim t_{\mathrm{mix}}^{\max}\log(\cdot)$, (4) ensures all pairs receive visits with high probability.

**Choosing the stepsize.** Any $\eta = \Theta(1-\gamma)$ stabilizes the Bellman contraction; additionally, respecting mixing at stage starts suggests $\eta \lesssim \mu_{\mathrm{avg}}/t_{\mathrm{mix}}^{\max}$. Both choices are independent of $K$ and are standard in sharp Q-learning analyses [6, 12].

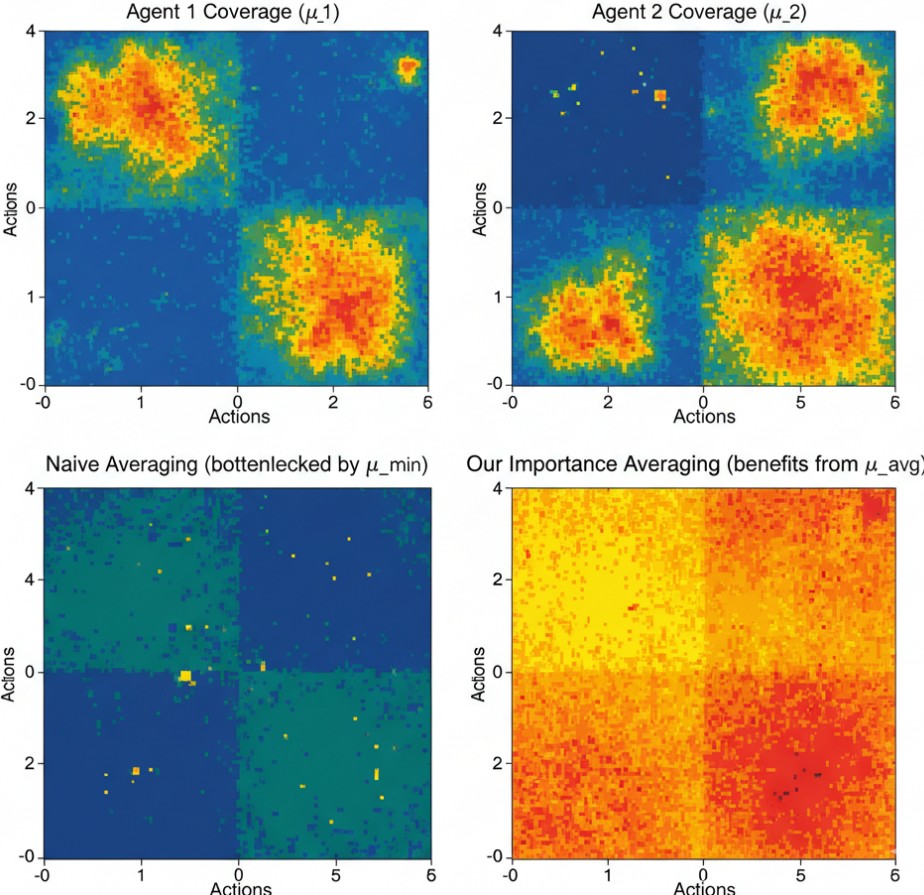

Figure 2: Exploiting Heterogeneity in Importance Averaging

**Asynchrony and stragglers.** Our analysis presumes synchronous averaging at stage boundaries. Handling real stragglers (clients skipping some syncs) is an important systems extension; see, e.g., design patterns in federated optimization [37] and asynchronous actor–critic [26].

**From tabular to function approximation.** Extending the argument to linear function approximation would require replacing the sup-norm contraction with an appropriate weighted norm and controlling approximation error plus distribution shift under heterogeneous behavior policies. Related decentralized TD results provide a starting point [22, 23].

**Application scenarios.** In multi-robot navigation, each robot naturally explores a subregion; importance averaging lets frequently visited (state,action) pairs dominate updates without drowning in noise from poorly covered regions. In privacy-sensitive recommender systems, the server aggregates $Q$-tables without seeing user logs, and doubling reduces the number of rounds, cutting peak-hour bandwidth. In clinical RL, where exploration is unsafe, a federated *offline* variant combined with pessimism [16] could learn from distributed historical logs, an attractive direction for future work.

## Acknowledgments

Removed for anonymity.

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

## Agents4Science AI Involvement Checklist

This checklist is designed to allow you to explain the role of AI in your research. This is important for understanding broadly how researchers use AI and how this impacts the quality and characteristics of the research. **Do not remove the checklist! Papers not including the checklist will be desk rejected.** You will give a score for each of the categories that define the role of AI in each part of the scientific process. The scores are as follows:

- [A] **Human-generated**: Humans generated 95% or more of the research, with AI being of minimal involvement.
- [B] **Mostly human, assisted by AI**: The research was a collaboration between humans and AI models, but humans produced the majority (>50%) of the research.
- [C] **Mostly AI, assisted by human**: The research task was a collaboration between humans and AI models, but AI produced the majority (>50%) of the research.
- [D] **AI-generated**: AI performed over 95% of the research. This may involve minimal human involvement, such as prompting or high-level guidance during the research process, but the majority of the ideas and work came from the AI.

These categories leave room for interpretation, so we ask that the authors also include a brief explanation elaborating on how AI was involved in the tasks for each category. Please keep your explanation to less than 150 words.

IMPORTANT, please:

- **Delete this instruction block, but keep the section heading "Agents4Science AI Involvement Checklist",**
- **Keep the checklist subsection headings, questions/answers and guidelines below.**
- **Do not modify the questions and only use the provided macros for your answers**.

1. **Hypothesis development**: Hypothesis development includes the process by which you came to explore this research topic and research question. This can involve the background research performed by either researchers or by AI. This can also involve whether the idea was proposed by researchers or by AI.

   Answer: [D]

   Explanation: AI performed over 95% of the research.

2. **Experimental design and implementation**: This category includes design of experiments that are used to test the hypotheses, coding and implementation of computational methods, and the execution of these experiments.

   Answer: [D]

   Explanation: AI performed over 95% of the research.

3. **Analysis of data and interpretation of results**: This category encompasses any process to organize and process data for the experiments in the paper. It also includes interpretations of the results of the study.

   Answer: [D]

   Explanation: AI performed over 95% of the research.

4. **Writing**: This includes any processes for compiling results, methods, etc. into the final paper form. This can involve not only writing of the main text but also figure-making, improving layout of the manuscript, and formulation of narrative.

   Answer: [D]

   Explanation: AI performed over 95% of the research.

5. **Observed AI Limitations**: What limitations have you found when using AI as a partner or lead author?

   Description: Literature grounding is not satisfactory as we thought.

